# Fast and selective organocatalytic ring-opening polymerization by fluorinated alcohol without a cocatalyst

Wei Zhao [1], Yanfeng Lv[1], Ji Li[1], Zihao Feng[1], Yonghao Ni[2] & Nikos Hadjichristidis[3]

Organocatalysis is an important branch of catalysis for various organic transformations and materials preparation. Polymerizations promoted by organic catalysts can produce polymeric materials without any metallic residues, providing charming materials for high-value and sensitive domains such as biomedical applications, microelectronic devices and food packaging. Herein, we describe a fluorinated alcohol based catalytic system for polypeptide synthesis via catalytic ring-opening polymerization (ROP) of $\alpha$-amino acid N-carboxyanhydride (NCA), fulfilling cocatalyst free, metal free, high rate and high selectivity. During polymerization, the fluorinated alcohol catalyst forms multiple dynamic hydrogen bonds with the initiator, monomer and propagating polymer chain. These cooperative hydrogen bonding interactions activate the NCA monomers and simultaneously protect the overactive initiator/propagating polymer chain-ends, which offers the whole polymerization with high activity and selectivity. Mechanistic studies indicate a monocomponent-multifunctional catalytic mode of fluorinated alcohol. This finding provides a metal free and fast approach to access well-defined polypeptides.

[1] College of Bioresources Chemical and Materials Engineering, Shaanxi University of Science and Technology, 710021 Xi'an, People's Republic of China. [2] Department of Chemical Engineering, University of New Brunswick, Fredericton, New Brunswick E3B 5A3, Canada. [3] KAUST Catalysis Center, Polymer Synthesis Laboratory, Physical Sciences and Engineering Division, King Abdullah University of Science and Technology (KAUST), Thuwal 23955, Saudi Arabia. Correspondence and requests for materials should be addressed to W.Z. (email: zhwgah1028@126.com)

Synthetic polypeptides, as mimics of natural analogues, is a unique family of bio-inspired biomaterials with broad bio-medical applications including controlled drug release, gene delivery, tissue engineering, and regenerative medicine[1,2]. Ring-opening polymerization (ROP) of α-amino acid N-carboxyanhydrides (NCAs) initiated by different amines and amines derivatives is considered to be the most common method for polypeptide synthesis. However, in the ROP of NCAs, the normal amine mechanism (NAM) and activated monomer mechanism (AMM), often compete with each other, which complicates the overall polymerization process, making it challenging to produce well-defined (co)polypeptides (see Supplementary fig. 1)[3,4]. In 1997, Deming reported the first example of controlled/living ROP of NCAs by using transition metal complexes instead of traditional primary amine as active species to control the monomer addition at polymer chain-end[5,6]. Since then, a few studies on controlled NCA polymerizations have been reported, by designing new initiators (e.g., Pt complexes[7], ammonium salts[8], silazane derivatives[9], rare earth metal complexes[10], amine-borane Lewis pairs[11], and Li complex[12]) or by exploring new reaction conditions for the traditional primary amine-initiated polymerizations (e.g., high vacuum[13,14], low temperature[15–17], and nitrogen flowing[18]). As for the ROP of NCA promoted by metal complexes, the metal residues in resultant polypeptides can cause toxicity concerns. An extra purification of the as-synthesized polymer such as dialysis is usually required. In regards to the NCA polymerizations initiated by organic initiators, low polymerization activities are usually observed, even with low monomer/initiator ratios[8,9,11], showing limitations in the synthesis of polypeptides with high molecular weights and/or for ROP of NCAs with low stabilities. Therefore, developing a fast and metal-free process for the synthesis of polypeptides is very necessary and promising.

Hydrogen-bonding is widely effective in biological systems, playing a crucial role in sustaining the elegant architecture/functionality of proteins and nucleic acids[19–21]. In an analogous manner to natural systems, organocatalysis utilizing hydrogen bonding to accelerate and control chemical reactions is of particular interest in organic chemistry and has become an active and vibrant research area in the past two decades[22–25]. Numerous classes of compounds such as binols[26–28], silane diols[29,30], squaramides[31,32], (thio)ureas[33,34], guanidine[35], and amidine functionalities[36] have been explored for various organic transformations. Furthermore, as an elegant alternative to organometallic and enzymatic catalysis, hydrogen-bonding organocatalysis has been a promising pathway for innovative polymer synthesis, especially for the preparation of metal-free

polymers required for advanced packaging, medical and micro-electronic applications[37–42].

In recent years, fluorinated alcohols have attracted extensive research interest due to their special properties such as lower boiling points and higher melting points than their non-fluorinated counterparts, high polarity, and strong hydrogen-bonding capacity. Because of these specificities, fluorinated alcohols, as solvents, cosolvents, and additives, have noteworthy applications[43,44]. For example, in biochemistry trifluoroethyl alcohol (TFE) and hexafluoroisopropyl alcohol (HFIP) are used to modify the conformation of proteins[45–47]. In organic chemistry, fluorinated alcohols are widely accepted as exceptional promotion media to perform organic reactions[43,48–50], including oxidation reactions with $H_2O_2$ or sodium hypochlorite[51–53], aza-Michael reaction[54], protection and deprotection of amine groups[55,56], cyclopropanation of alkenes[57], and oxirane ring-opening[58,59]. In polymer chemistry, fluorinated alcohols are effective solvents to control the stereospecificity of atom transfer radical polymerization (ATRP)[60–63] and conventional radical polymerization[64,65]. So far, fluorinated alcohols, as small molecule metal free catalysts, rather than reaction mediums for polymer synthesis, are seldom explored. The only attempt refers to a fluorinated alchohol/(−)-sparteine bicomponent catalytic system developed by Hedrick et al. for ROP of cyclic esters, in which a fluorinated alcohol is used to activate the monomer and an organic base cocatalyst (−)-sparteine to activate the initiating/propagating alcohol. Without (−)-sparteine, the polymerization cannot happen (Fig. 1a)[66] To the best of our knowledge, so far there has been no report of utilizing a fluorinated alcohol as a catalyst for polymerization in the absence of base cocatalyst.

In this contribution, we show that 1,3-bis(2-hydroxyhexafluoroisopropyl)benzene (1,3-Bis-HFAB), a small fluorinated alcohol, without the assistance of a Lewis base cocatalyst, can catalyze a fast and selective ROP of α-amino acid N-carboxyanhydride (NCA), resulting in well-defined polypeptides. Different from the bicomponent-bifunctional catalytic process of fluorinated alchohol/(−)-sparteine system for ROP of cyclic esters[66], the 1,3-Bis-HFAB catalyst doesn't require a Lewis base cocatalyst, and it is defined as a monocomponent-multifunctional catalytic process for ROP of NCA (Fig. 1b). During polymerization, 1,3-Bis-HFAB can form multiple dynamic hydrogen bondings with initiator, monomer and propagating polymer chain-end, moving between them. These cooperative hydrogen-bonding interactions activate the monomer and simultaneously protect the overactive initiator/polymer chain from side reactions, offering high reaction rates and selectivity in the polymerization (Fig. 2). This finding not only represents a nonconventional catalysis

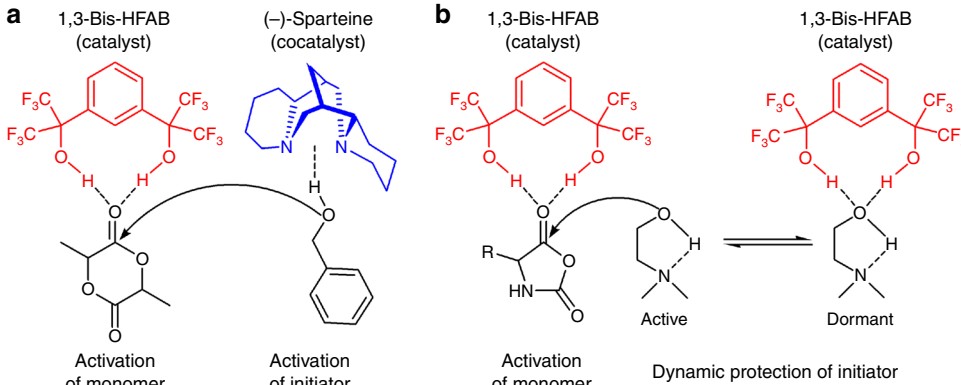

**Fig. 1** Two different fluorinated alcohol based organocatalytic systems for ROP. **a** the bicomponent-bifunctional fluorinated alcohol/(−)-sparteine catalytic system for ROP of cyclic esters in which cocatalyst base, (−)-sparteine, is required. **b** the concept of monocomponent-multifunctional fluorinated alcohol catalytic system to be developed for ROP of NCA in this study

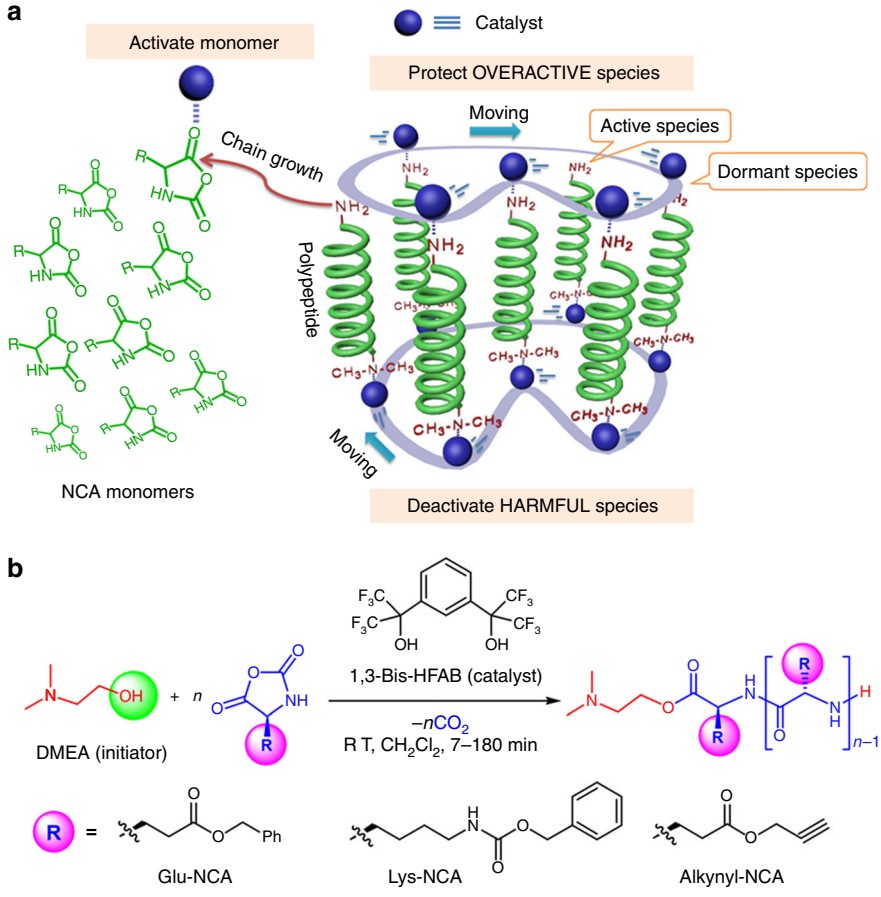

**Fig. 2** Organocatalytic ROP of $\alpha$-amino acid *N*-carboxyanhydride (NCA) by fluorinated alcohol. **a** schematic diagram of the roles of fluorinated alcohol catalyst played during polymerization; **b** (co)polymerization of Glu-NCA, Lys-NCA, and Alkynyl-NCA initiated by DMEA in the presence of fluorinated alcohol catalyst 1,3-Bis-HFAB

methodology of fluorinated alcohols but also provides a pathway for polypeptides synthesis, fulfilling metal free, high activity, and high selectivity. Other merits of this strategy are: (1) the 1,3-Bis-HFAB catalyst and the aminoalcohol initiator used in this study are all commercially available organic chemicals, (2) the catalyst can be easily removed from final polypeptide after polymerization, which provides a facile way to get clean polypeptides for researchers who focus on the functions and applications of polypeptides but have backgrounds other than organic chemistry.

## Results

**Ring-opening polymerization of Glu-NCA promoted by 1,3-Bis-HFAB.** The catalytic performance of 1,3-Bis-HFAB for ROP of Glu-NCA was evaluated by using a small aminoalcohol molecule, *N,N*-dimethylethanolamine (DMEA), as the initiator. The catalyst 1,3-Bis-HFAB and the initiator DMEA were dissolved in dichloromethane, followed by introduction of a dichloromethane solution of Glu-NCA monomer at room temperature. Under these conditions, the polymerization reaction proceeded not only at a high rate but also high selectivity, with negligible side reactions, such as chain-transfer and chain-termination. As shown in Table 1, for a monomer-to-initiator ratio of 120 ([Glu-NCA]/[DMEA] = 120) and 1 equiv. 1,3-HFAB in dichloromethane, the NCA monomer was completely converted to polypeptide in just 7 min at room temperature, producing polypeptide with a narrow molecular weight distribution Đ (Đ = $M_w/M_n$ = 1.07, $M_{n,SEC}$ = 4.5 × 10$^4$; Table 1, run 1). When the monomer-to-initiator ratio ([Glu-NCA]/[DMEA]) was gradually increased from 120 to 1920 by decreasing the

concentration of initiator [DMEA] ([Glu-NCA] and [1,3-Bis-HFAB] were fixed, Table 1, runs 1–5), the molecular weights ($M_{n,SEC}$) of polypeptides increased linearly as a function of [Glu-NCA]/[DMEA] ratio and the molecular weight distributions remained narrow (Đ = 1.05–1.07), indicative of the living nature of the polymerization (Fig. 3a, b). The end-group analysis of a low molecular weight sample, prepared under analogous conditions, by $^1$H NMR spectroscopy gave further support of the living nature of the polymerization. The resonances of DMEA were evident in the $^1$H NMR spectrum and the $M_{n,NMR}$ calculated based on $^1$H NMR spectrum was close to the theoretical value (Fig. 3c). The chemical structure of the obtained polypeptide was supported further by matrix-assisted laser desorption/ionization time-of-flight mass spectroscopy (MALDI-TOF MS). In the MALDI-TOF mass spectrum of oligomeric polypeptide, only one population was found (Fig. 3d). Based on the *m/z* value, a formula of $\{(CH_3)_2NCH_2CH_2O-[COCH(CH_2CH_2COOCH_2C_6H_5)NH]_n-COCF_3\}H^+$ ($n$ = 7–23) was established, which is in good agreement with an oligomeric polypeptide having −NH$_2$ and DMEA end-groups ($M_{MS}$ = 3450, Đ$_{MS}$ = 1.05). These results suggest that with DMEA as an initiator, the NCA monomers can undergo rapid and selective polymerization, giving polypeptides with well-defined molecular weight and low Đ.

Besides possessing a high selectivity for propagating chains, the ROP of Glu-NCA promoted by 1,3-Bis-HFAB exhibits high activities. When the [Glu-NCA]/[DMEA] ratio was changed from 120 to 960 by varying the initiator concentration [DMEA] (Table 1, runs 1–4), all polymerizations could be completed in 70 min. Even at a high [Glu-NCA]/[DMEA] ratio up to 1920, the

**Table 1 Living polymerization of NCAs promoted by 1,3-Bis-HFAB**

| Run[a] | Monomer | [M]/[I]/[Cat.] | [Cat.] (mol%) | time (min) | $M_{n,calcd} \times 10^{-4}$ [b] | $M_{n,SEC} \times 10^{-4}$ [c] | Đ[c] |
|---|---|---|---|---|---|---|---|
| 1 | Glu-NCA | 120/1/1 | 0.8 | 7 | 2.63 | 4.57 | 1.07 |
| 2 | Glu-NCA | 120/0.5/1 | 0.8 | 13 | 5.26 | 7.43 | 1.08 |
| 3 | Glu-NCA | 120/0.25/1 | 0.8 | 25 | 10.52 | 12.72 | 1.04 |
| 4 | Glu-NCA | 120/0.125/1 | 0.8 | 70 | 21.04 | 20.33 | 1.04 |
| 5 | Glu-NCA | 120/0.0625/1 | 0.8 | 180 | 37.88 | 35.87 | 1.05 |
| 6 | Glu-NCA | 120/1/2 | 1.6 | 8.5 | 2.63 | 4.16 | 1.06 |
| 7 | Glu-NCA | 120/1/4 | 3.2 | 10.5 | 2.63 | 3.68 | 1.07 |
| 8 | Glu-NCA | 120/1/6 | 4.8 | 12.5 | 2.63 | 3.49 | 1.05 |
| 9 | Glu-NCA | 120/1/8 | 6.4 | 15 | 2.63 | 3.43 | 1.04 |
| 10 | Glu-NCA | 120/1/10 | 8 | 16 | 2.63 | 3.15 | 1.04 |
| 11 | Lys-NCA | 120/1/1 | 0.8 | 9 | 3.15 | 3.35 | 1.06 |
| 12 | Lys-NCA | 120/0.5/1 | 0.8 | 16 | 6.30 | 6.65 | 1.08 |
| 13 | Lys-NCA | 120/0.25/1 | 0.8 | 35 | 12.59 | 13.01 | 1.07 |
| 14 | Alkynyl-NCA | 120/1/1 | 0.8 | 8 | 2.01 | 2.31 | 1.05 |
| 15 | Alkynyl-NCA | 120/0.5/1 | 0.8 | 15 | 4.01 | 4.21 | 1.07 |
| 16[d] | Glu/Lys-NCA | (60 + 60)/1/1 | 0.8 | 5 + 5 | 2.89 | 3.08 | 1.08 |
| 17[e] | Glu/Lys-NCA | (60 + 60)/1/1 | 0.8 | 10 | 2.89 | 3.16 | 1.05 |
| 18[f] | Glu/Alkynyl-NCA | (60 + 60)/1/1 | 0.8 | 5 + 5 | 2.32 | 2.48 | 1.07 |
| 19[g] | Glu/Alkynyl-NCA | (60 + 60)/1/1 | 0.8 | 5 + 5 | 2.32 | 2.61 | 1.04 |

[a]Polymerization was performed at 25 °C with $[NCA]_0 = 0.19$ M. In situ FT-IR was used to determine the conversion of NCA by analysing the intensity of the NCA anhydride absorption band at 1792 cm$^{-1}$ and 100% monomer conversion was achieved except run 5 (90%)
[b]Calculated by [M]/[I] × ($M_{NCA}$ − 44) × Conv
[c]Determined by size-exclusion chromatography (SEC) combined with multiangle light scattering (MALS), viscometry (VISC), and differential refractive index (DRI) triple detection in 0.1 M LiBr in DMF at 60 °C
[d]Sequential polymerization of Glu-NCA and Lys-NCA
[e]Random copolymerization of Glu-NCA and Lys-NCA
[f]Sequential polymerization of Glu-NCA and Alkynyl-NCA
[g]Random copolymerization of Glu-NCA and Alkynyl-NCA

polymerization proceeded at a considerably high rate, and initiator loading could be as low as 0.01 mol% (vs. [Glu-NCA]) without significant loss of activity. Ninety percent monomer conversion was achieved in 3 h, producing a polypeptide with narrow molecular weight distribution and a molecular weight close to the theoretical value (Đ = $M_w/M_n$ = 1.05, $M_{n,SEC}$ = 35.87 × 10$^4$; Table 1, run 5).

**Kinetics analysis**. We further studied the reaction kinetics by using in situ FT-IR technique (Fig. 4). At fixed monomer and initiator concentrations ([Glu-NCA]$_0$ = 0.19 M and [DMEA]$_0$ = 4.0 × 10$^{-4}$ M), the conversion of Glu-NCA with time was monitored by in situ FT-IR at various catalyst concentrations ([1,3-Bis-HFAB]$_0$ = 1.7, 3.3, 5.0, 6.7, and 17 mol% vs. monomer) until full conversion was reached. Sampling was from 2800 cm$^{-1}$ to 650 cm$^{-1}$ at 8 wavenumber resolution and the automatic sampling interval was 10 s, which guaranteed the real-time data acquisition of polymerization (Fig. 4a, b). The polymerization possesses classic two-stage propagation, which is consistent with the literature reports. The ROP of Glu-NCA in a number of solvents is shown to proceed at two successive rates following a relatively rapid initiation. The two-stage propagation is due to the transition of polypeptide chain from $\beta$ (non-helical, low DP) to $\alpha$ (helical, high DP) configuration during growth[67,68]. The rate constant for the second propagation step is at least several times larger than that for the first. Thus, the whole polymerization rate is largely dependent on the second stage propagation. As shown in Fig. 4, a first-order dependence of polymerization rate on monomer concentration was observed at the second propagation in each experiment. The corresponding apparent rate constants ($k_{app}$ = 1.50 × 10$^{-3}$ s$^{-1}$, 1.20 × 10$^{-3}$ s$^{-1}$, 9.98 × 10$^{-4}$ s$^{-1}$, 9.29 × 10$^{-4}$ s$^{-1}$, and 6.29 × 10$^{-4}$ s$^{-1}$) were obtained from the semilogarithmic plots Ln([M]$_0$/[M]$_t$) = $k_{app}$ t. The plot of Ln$k_{app}$ versus Ln[1,3-Bis-HFAB]$_0$ is linear, and the gradient of the best-fit line is −0.3785. Therefore, the kinetic order in catalyst is equal to −0.40 (within experimental errors). The minus order

in catalyst indicates that high loading amount of catalyst in the system can cause a decrease of the polymerization rate.

The dependence of polymerization rate on initiator concentration was also determined (Fig. 4c, d). At fixed monomer and catalyst concentrations ([Glu-NCA]$_0$ = 0.19 M and [1,3-Bis-HFAB]$_0$ = 1.58 × 10$^{-3}$M), the conversion of Glu-NCA with time was monitored at various initiator concentrations ([DMEA]$_0$/ [1,3-Bis-HFAB]$_0$ = 1, 0.5, 0.25, 0.125, and 0.0625). Again, the consumption of the monomer follows the first-order kinetics. Hence the kinetic order in initiator is 1.45 (within experimental errors). As a result, the polymerization rate law is $R_P = -d[M]/dt = k_p[1,3\text{-Bis-HFAB}]^{-0.40}[DMEA]^{1.45}[NCA]$.

**NMR study of interactions between catalyst, monomer, and propagating chain**. To understand the high activity and selectivity of polymerization at a molecular level, we investigated the interactions of 1,3-Bis-HFAB with Glu-NCA monomer, initiator, and propagating chains by using nuclear magnetic resonance (NMR) spectroscopy (Fig. 5). The $^1$H NMR comparative study of 1,3-Bis-HFAB, Glu-NCA and 1,3-Bis-HFAB/Glu-NCA mixture revealed that 1,3-Bis-HFAB didn't induce the ROP of NCA and acted as a hydrogen-bonding donor to Glu-NCA. When 1,3-Bis-HFAB was mixed with equivalent Glu-NCA monomer in anhydrous DCM-d$_2$ and monitored at room temperature for 24 h, no evidence of polymer was observed in the $^1$H NMR spectrum. Due to the hydrogen-bonding interaction between Glu-NCA and 1,3-Bis-HFAB, the -OH of 1,3-Bis-HFAB shifted downfield from 3.66 to 3.92 ppm in the $^1$H NMR spectrum. Only one hydroxyl proton signal was observed, indicative of a fast equilibrium on the NMR time scale (Fig. 5a and Supplementary fig. 2). In addition, the −C$F_3$ of 1,3-Bis-HFAB also showed a downfield chemical shift (from −75.92 to −75.87ppm) in the $^{19}$F NMR spectrum (Fig. 5b and Supplementary fig. 3). The hydrogen-bonding interactions between 1,3-Bis-HFAB and the oxygen of carbonyl can dramatically enhance the activity of the carbonyl group.

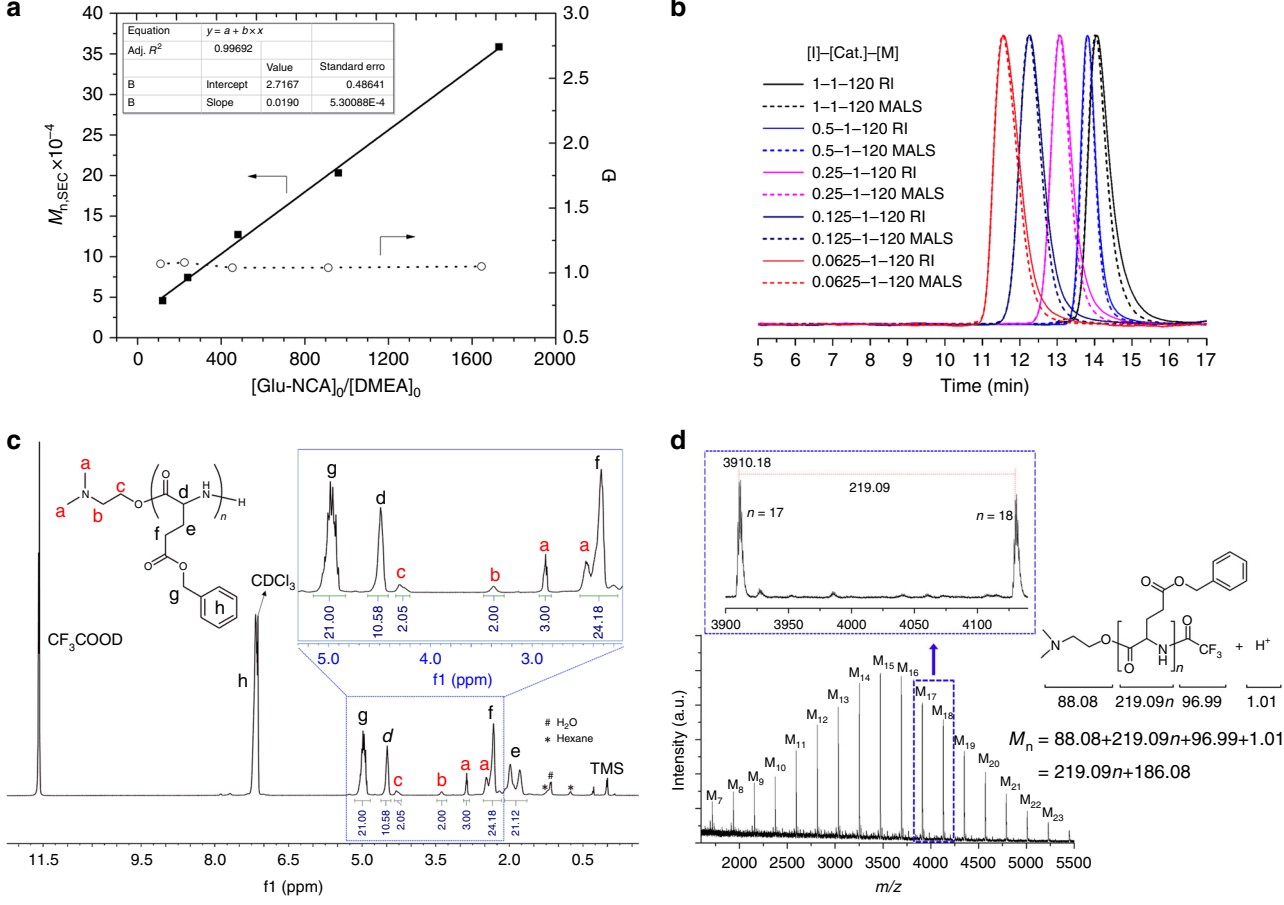

**Fig. 3** Characterization of polypeptides synthesized from ROP of Glu-NCA initiated by DMEA in the presence of 1,3-Bis-HFAB catalyst. **a** plots of [Glu-NCA]$_0$/[DMEA]$_0$ vs $M_{n,SEC}$ and [Glu-NCA]$_0$/[DMEA]$_0$ vs Đ; **b** SEC traces of obtained polypeptides (Table 1, runs 1–5, solid and dash lines represents RI and MALS responses, respectively); **c** $^1$H NMR spectrum of PBLG$_{10}$ (500 M, 25 °C, CDCl$_3$/CF$_3$COOD = 2, the $M_{n,calcd}$ is 2192 and the $M_{n,NMR}$ calculated based on the integral ratio of H$^g$/H$^b$ is 2302; **d** MALDI-TOF MS spectrum of PBLG$_{15}$ (the chain-end was capped with trifluoroacetyl group by using trifluoroacetic anhydride). (Source data are provided as a Source Data file)

The initiator, DMEA, contains a primary alcohol and a tertiary amine. According to previous literature, the tertiary amine can trigger uncontrolled NCA polymerization via an activated monomer mechanism, resulting in ill-defined polypeptide[3,4,69]. However, the end-group analysis of polypeptide by $^1$H NMR and MALDI-TOF in the present study shows that the tertiary amine of DMEA is totally silent during polymerization (Fig. 3c, d). In order to figure out the reason, we protected the hydroxyl group of DMEA with tert-butyldimethylsilane (resulting in DMEA-TMS') and investigated the effect of 1,3-Bis-HFAB on the tertiary amine of DMEA. In the $^1$H NMR spectrum of 1,3-Bis-HFAB with equivalent DMEA-TMS', the –OH of 1,3-Bis-HFAB shifted downfield from 3.66 to 6.89 ppm and the signals of DMEA-TMS' (H$^e$, H$^g$, and H$^f$) also shifted downfield slightly (Fig. 5a and Supplementary Figs. 4 and 5). In the $^{19}$F NMR spectrum, the fluorine resonance of –CF$_3$ in 1,3-Bis-HFAB shifted downfield from −75.92 to −75.61ppm (Fig. 5b and Supplementary fig. 3). The large change in chemical shift of the –OH of 1,3-Bis-HFAB (Δ = 3.23) suggests that a strong hydrogen-bonding interaction takes place between 1,3-Bis-HFAB and the tertiary amine of DMEA. We further found that, DMEA-TMS', in the absence of 1,3-Bis-HFAB, could trigger the ROP of Glu-NCA via activated monomer mechanism. Kinetics studies via in situ FI-IR showed that the polymerization was very slow (24 h, 92% monomer conversion) and a first-order dependence of polymerization rate on monomer concentration was not observed, indicating a slow

initiation of the polymerization. The SEC demonstrated that the resultant polypeptide possessed an ultrahigh molecular weight ($M_{n,SEC} = 2.67 \times 10^5$, ten times that of the calculated molecular weight $M_{n,calcd} = 2.63 \times 10^4$) and a very broad molecular weight distribution (Đ = 1.63) (Supplementary fig. 6). However, in the presence of 1,3-Bis-HFAB, DMEA-TMS' cannot initiate the polymerization, which indicates that the strong hydrogen-bonding interaction can neutralize the initiation capability of the tertiary amine to trigger polymerization. So, the primary alcohol is the only initiating site, which is consistent with the conclusion of end-group analysis by $^1$H NMR and MALDI-TOF MS spectrum (Fig. 3c, d). If the tertiary amine in DMEA was involved in the initiation of polymerization, an activated monomer mechanism of NCA polymerization could be triggered and the initiator would not quantitatively remain at the polymer chain-ends after polymerization, which would give much higher experimental $M_n$ than the theoretical one. This is further solidified by studying the ROP of Glu-NCA initiated by N, N-dimethyl-1,2-ethanediamine (DMEDA, (CH$_3$)$_2$NCH$_2$CH$_2$NH$_2$) in the absence of 1,3-bis-HFAB (conditions: [Glu-NCA]/[DMEDA] = 100, [Glu-NCA]$_0$ = 0.19 M, 25 °C). DMEDA has both tertiary amine and primary amine in its molecular structure and can trigger the NCA polymerization via both mechanisms. The SEC characterization demonstrated that the resultant polypeptide possessed a bimodal molecular weight distribution (Đ = 1.38) and the experimental molecular weight

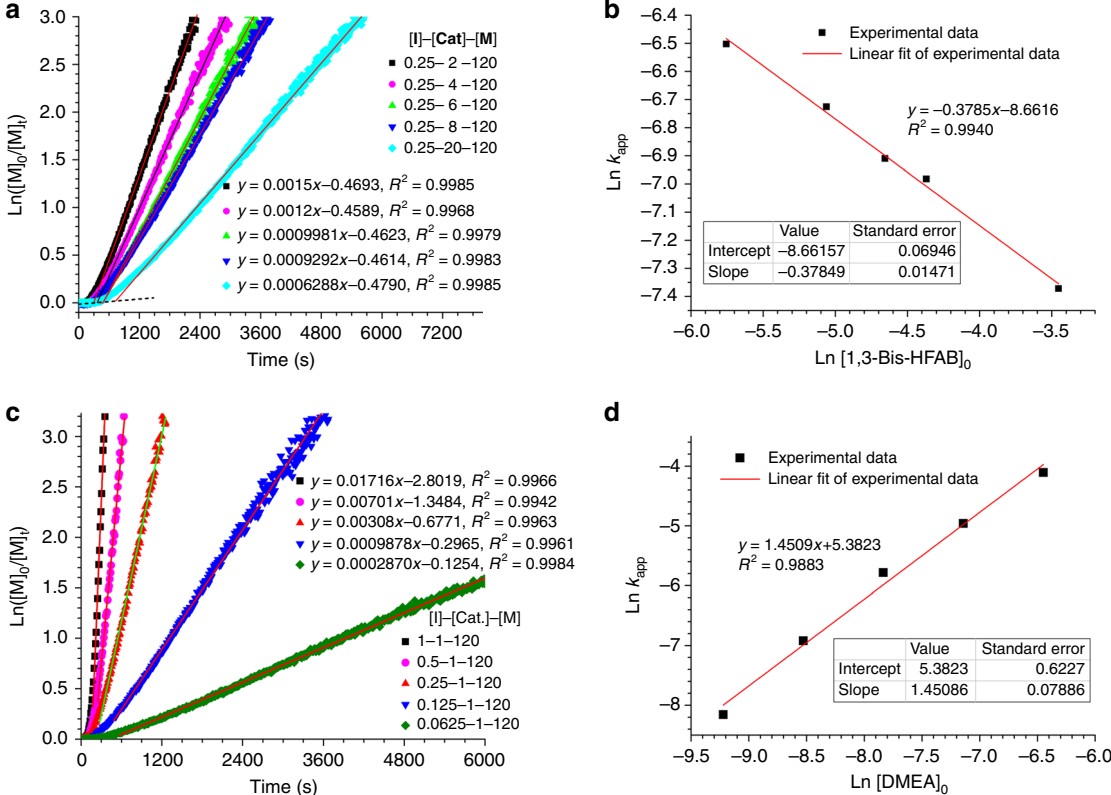

**Fig. 4** Kinetics study of Glu-NCA polymerization catalyzed by 1,3-Bis-HFAB at room temperature in $CH_2Cl_2$. **a, b** $[1,3\text{-Bis-HFAB}]_0 = 1.7\text{–}17$ mol% vs. $[\text{Glu-NCA}]_0$, $[\text{DMEA}]_0 = 4 \times 10^{-4}$ M and $[\text{Glu-NCA}]_0 = 0.19$ M; **c, d** $[1,3\text{-Bis-HFAB}]_0 = 1.58 \times 10^{-3}$ M, $[\text{DMEA}]_0 = 0.05\text{–}0.8$ mol% vs. $[\text{Glu-NCA}]_0$ and $[\text{Glu-NCA}]_0 = 0.19$ M. (Source data are provided as a Source Data file)

$(M_{n,SEC} = 4.71 \times 10^4)$ is more than two times higher than the theoretical molecular weight $(M_{n,calcd} = 2.19 \times 10^4)$ (Supplementary fig. 7).

Since the polymer chain-end will be primary amine group after initiating by the hydroxyl group of DMEA, we further investigated the interaction of 1,3-Bis-HFAB with polymer chain-end by using 2-amino-2-phenylacetate (MAP), as a model molecule. The model molecule has a structure similar to the polymer chain end. The $^1H$ NMR results revealed that hydrogen bonding was also formed between 1,3-Bis-HFAB and chain-end primary amine group. In the presence of 1,3-Bis-HFAB, the signal of the primary amine group in MAP shifted downfield from 1.84 to 4.21 ppm and the signal of the hydroxyl protons in 1,3-Bis-HFAB shifted downfield from 3.66 to 4.21 ppm (Fig. 5a and Supplementary fig. 8). In the $^{19}F$ NMR spectrum, the fluorine resonance of $-CF_3$ in 1,3-Bis-HFAB shifted downfield from $-75.92$ to $-75.68$ ppm (Fig. 5b and Supplementary fig. 3). These results indicate that the nucleophilic character of the chain-end primary amine group decreases upon formation of hydrogen-bonding with 1,3-Bis-HFAB. By increasing of the loading amount of 1,3-Bis-HFAB in the system, the polymerization slowed down, which is consistent with the kinetic study results. However, under identical reaction conditions, polymerization experiments with different 1,3-Bis-HFAB loadings resulted in polymers with similar targeted $M_n$ and very low Đ (Table 1, runs 7–10). This suggests that 1,3-Bis-HFAB transforms the active species to dormant species without killing them and that a quick and reversible exchange exists between the free and dormant primary amine. This faster (vs. the propagation) and reversible exchange gives all amine groups an equal chance to trigger the polymerization. It seems that the catalyst is moving quickly between different

chain-end primary amine groups. The dormant/active site exchange of primary amine group is further supported by the result from in situ FT-IR monitoring of polymerization (Fig. 6). The rate of polymerization slowed down after adding excess 1,3-Bis-HFAB into the system during the polymerization, but the final polymer possessed $M_{n,SEC}$ of $3.01 \times 10^4$ and very low Đ of 1.05 (the SEC trace see Supplementary fig. 9). If the active species would be killed by 1,3-Bis-HFAB during polymerization, the polymerization would slow down and the $M_n$ of obtained polymer would be much higher than the targeted one $(2.63 \times 10^4)$.

According to previous reports in the literature, the chain-end primary amine is prone to side reactions during polymerization, which make it challenging to synthesize polymers with complex architectures like block copolymers[3,4]. Therefore, the hydrogen bonding formed between 1,3-Bis-HFAB and chain-end primary amine can provide a dynamical protection of the overactive chain-end amine and suppress the occurrence of side reactions in system. A control experiment was carried out by using a primary amine, n-hexylamine (HA) as initiator for the ROP of Glu-NCA, in the absence of 1,3-Bis-HFAB (conditions: [Glu-NCA]/[HA] = 120, $[\text{Glu-NCA}]_0 = 0.19$ M, 25 °C, $CH_2Cl_2$). The results showed that the polymerization was very slow, forming polypeptide with a broad molecular weight distribution (4.5 h, 98% conv., $M_{n,SEC} = 3.23 \times 10^4$, Đ = 1.17, $M_{n,calcd} = 2.57 \times 10^4$). A further kinetics study via in situ FT-IR method showed that the polymerization process did not have a constant reaction rate (Supplementary fig. 10). When the monomer conversion was above 82%, the polymerization slowed down and deviated from the living polymerization due to side reactions. When the monomer/initiator ratio increased from 120 to 240, the results were even worse. Only 68% monomer were converted to polymer

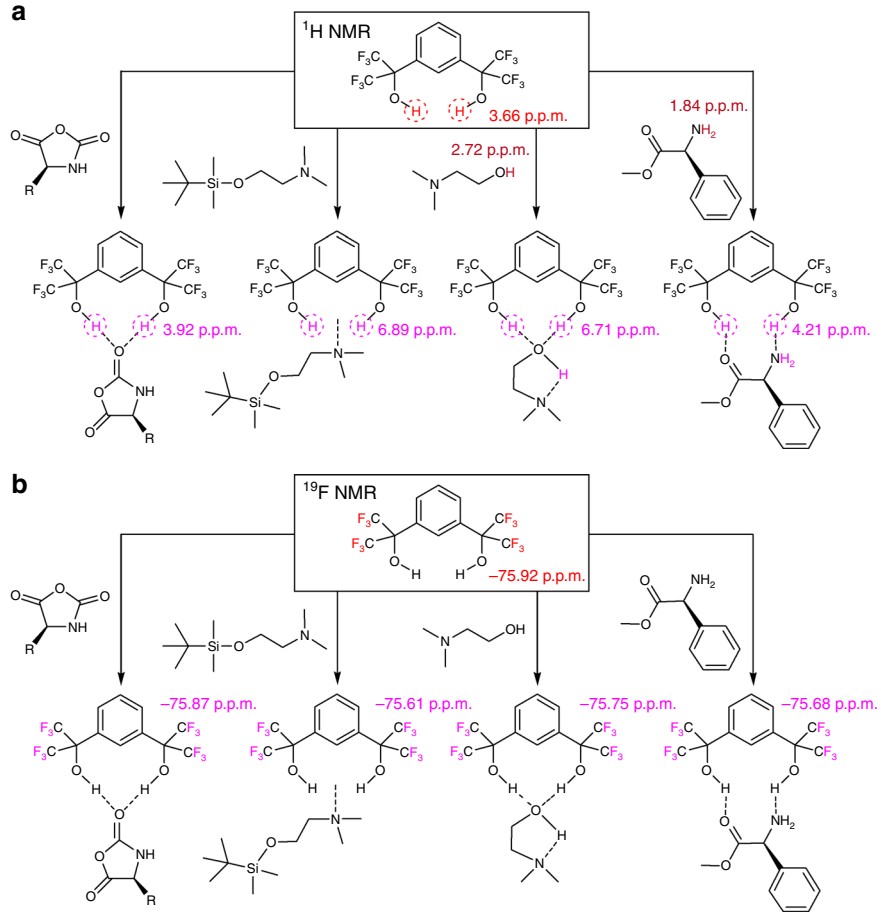

**Fig. 5** Chemical shift changes of hydroxyl groups and fluorine atoms in 1,3-Bis-HFAB. **a** observed in the NMR spectra of 1,3-Bis-HFAB with Glu-NCA, DMEA-TMS', DMEA, and MAP, respectively; **b** observed in $^{19}F$ NMR spectra (the comparative spectra see supplementary Figs. 2–5 and 8) (Source data are provided as a Source Data file)

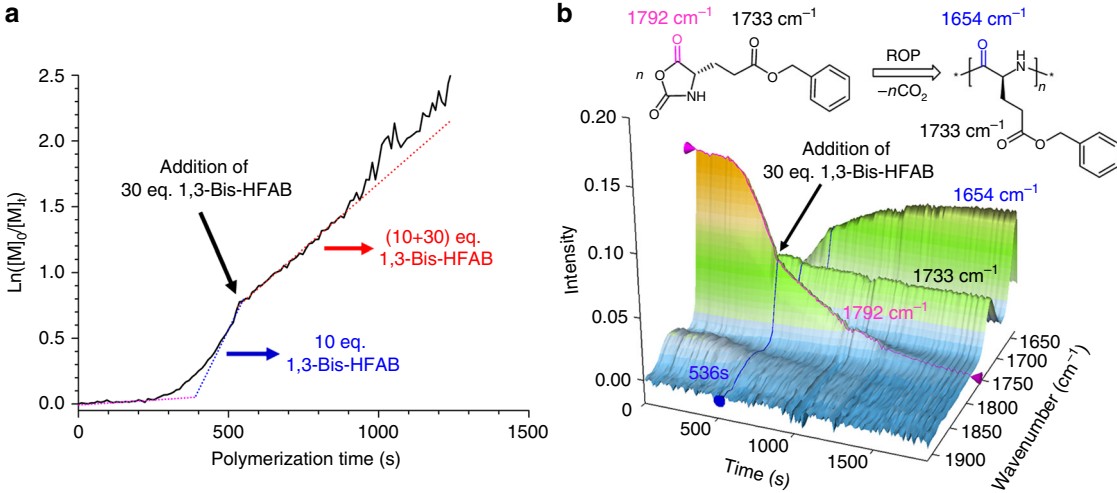

**Fig. 6** In situ FT-IR monitoring of polymerization. **a** plot of $Ln([M]_0/([M]_t)$ vs. polymerization time (conditions: $[Glu-NCA]_0 = 0.19$ M, $[Glu-NCA]_0/[1,3-Bis-HFAB]_0/[DMEA]_0 = 120/(10 + 30)/1$, 25 °C, $CH_2Cl_2$); **b** corresponding 3D kinetic behavior profile from in situ FT-IR (Source data are provided as a Source Data file)

in 9 h. The resultant polymer showed a bimodal SEC trace (with a shoulder peak) and a higher molecular weight ($M_{n,SEC-} = 6.10 \times 10^4$) than the theoretical molecular weight ($M_{n,calcd} = 3.58 \times 10^4$). Also, the monomer could not be completely converted to polypeptides even with a very long reaction time (1 day).

**Computational study**. To further understand the catalytic activity of 1,3-Bis-HFAB, we also carried out a computational study and investigated the interactions of 1,3-Bis-HFAB with Glu-NCA monomer, DMEA initiator, and propagating chain. The calculation was performed at the GGA/PBE/DNP level of

theory. All of the constructed structures are fully optimized and electrostatic potential (ESP)-fitting charges are derived from the DFT calculation. The lowest-energy structure of 1,3-Bis-HFAB/Glu-NCA complex ($\Delta E = -57.61$ kcal/mol) demonstrated that the –OH of 1,3-Bis-HFAB formed hydrogen-bonding with the C(5)=O of NCA monomer. The charges assigned to the two hydrogen atoms of –OH groups in 1,3-Bis-HFAB decreased from 0.393/0.390 to 0.380/0.238 and the charge of the oxygen atom of NCA carbonyl group decreased from $-0.350$ to $-0.389$ (Supplementary fig. 11). The minimum energy structure of 1,3-Bis-HFAB/DMEA complex ($\Delta E = -58.73$ kcal/mol) showed that the charge of the oxygen atom of DMEA increased from $-0.497$ to $-0.229$ after forming hydrogen-bonding with 1,3-Bis-HFAB, rendering the hydroxyl group of DMEA in a dormant state without nucleophilicity (Supplementary fig. 11). The complex structures of 1,3-Bis-HFAB/DMEA with one incorporated Glu-NCA monomer were also optimized ($\Delta E = -80.63$ kcal/mol). When the first Glu-NCA monomer was ring-opened by DMEA, 1,3-Bis-HFAB would form hydrogen-bondings with the tertiary amine from DMEA and the primary amine derived from the ring-opened monomer. The charge of nitrogen atom of –N(CH$_3$)$_2$ increased from $-0.126$ to 0.318 and lost nucleophilicity. The charge of nitrogen atom of –NH$_2$ increased from $-0.867$ to $-0.713$, indicative of a decreased nucleophilicity of –NH$_2$ (Supplementary fig. 11). When the secondary monomer is incorporated, the minimum energy structure ($\Delta E = -59.93$ kcal/mol) showed that 1,3-Bis-HFAB was still bound to the tertiary amine of DMEA. The charge of nitrogen atom of tertiary amine is 0.197 and has no nucleophilicity. The above is consistent with the polymerization result that the tertiary amine is not capable of triggering the polymerization in the presence of 1,3-Bis-HFAB after opening the first cyclic NCA monomer molecule.

**Proposed polymerization mechanism**. On the basis of the results from polymerization kinetics, NMR experiments and computational calculation, a plausible reaction pathway is shown in Scheme 3. As shown, multiple hydrogen-bonding equilibriums involving NCA monomer, initiator (DMEA), catalyst (1,3-Bis-HFAB) and propagating polymer chain exist. We postulate that the catalysis proceeds by simultaneous activation of the carbonyl of a NCA monomer and dynamic protection of the polymer chain-end amine group via hydrogen bonding to 1,3-Bis-HFAB. Nucleophilic ring-opening of the NCA leads to propagation, whereby the ring-opened NCA forms the propagating amine for the subsequent addition of monomer (Fig. 7). During polymerization, 1,3-Bis-HFAB moves between monomer and polymer chain-ends and plays triple roles through donating hydrogen-bonding: (1) increasing initiation and propagation rates by activating monomer; (2) dynamically protecting the overactive propagating primary amine from side reactions; and (3) totally silencing the tertiary amine of initiator after the initiation step. Although the high loading of 1,3-Bis-HFAB will increase the concentration of dormant species in the system and consequently cause a slight decrease of polymerization activity, this negative side effect can be partly compensated for by the activation of NCA monomer by 1,3-Bis-HFAB. That is why the polymerization can still proceed at a high rate.

**NCA monomer extension and catalyst residue analysis**. In order to explore the generality of this strategy for polypeptide synthesis, we also investigated the homopolymerizations of Lys-NCA and Alkynyl-NCA and the copolymerizations of Glu-NCA with Lys-NCA and Alkynyl-NCA (Table 1, runs 11–19). The polymerization results showed that the method is not monomer specific and can be used to prepare a series of well-defined random/block (co)polypeptides. The polymerization activity is different, depending on the monomer's structure, but all polymerization of these monomers can be completed in 10 min at room temperature with a monomer/initiator ratio of 120. We also checked the 1,3-Bis-HFAB catalyst residue in the polymer samples by using $^1$H NMR, $^{19}$F NMR, and EDS. We purposely prepared a high molecular weight polypeptide sample from polymerization with high loading of catalyst (conditions: [Glu-NCA]/[DMEA]/[1,3-Bis-HFAB] = 120/0.25/10, [Glu-NCA] = 0.19 M, 25 °C). When the polymerization is complete, and the resultant polypeptide was precipitated using methanol, a further two-step washing process of the precipitated polypeptide with methanol can totally remove the catalyst from the final polymer. The $^1$H NMR, $^{19}$F NMR, and EDS spectra show no 1,3-Bis-HFAB catalyst residue in the final polypeptide (Supplementary figs. 12–21). Furthermore, we also evaluated the cytotoxicity of the final polypeptide by doing in vitro cell experiments as followers. Briefly, human HepG2 cells were seeded in 96-well plates with a density of $5 \times 10^3$ cells per well and incubated overnight. Subsequently, the cells were incubated for 48 h with various concentrations of the extract of the final polypeptides (the preparation of extract of polypeptide follows international standard ISO 10993-12:2012). Afterwards, the cytotoxicity was evaluated by adding CCK-8 solution to each well of the plate. The cell viabilities with the extracts of polypeptide exceeded 95% (Supplementary fig. 22). Also, by using live/dead cell staining technique, stained live and dead cells can be visualized by fluorescence microscopy. From fluorescence microscopy images, we easily found that the cells viabilities with polypeptide extracts were comparable to the blank sample (Supplementary fig. 23). These results suggest that the final polypeptide does not show any cytotoxicity to HepG2 cells.

## Discussion

In this study, we report a nonconventional catalysis methodology of fluorinated alcohol and successfully applied it to controlled synthesis of polypeptides. The methodology has some unique advantages over existing ones; free of cocatalyst, free of metal, free of high-boiling point solvent, high activity, and selectivity at room temperature. In addition, the 1,3-Bis-HFAB catalyst and the aminoalcohol initiator used in this study are all commercially available organic chemicals and the catalyst can be easily removed from final polypeptide after polymerization. These characteristics are crucial for the high-value-added applications of polypeptides in microelectronic and medical fields. Further development of this methodology will be targeted towards (1) expanding NCA monomers, (2) functionalizing polypeptides and polypeptide hybrid copolymers by designing new initiators.

## Methods

**Measurements**. $^1$H, $^{19}$F, and $^{13}$C NMR spectra were recorded on a Bruker AV500 spectrometer (FT, 500 MHz for $^1$H and $^{19}$F; 125 MHz for $^{13}$C). Chemical shifts are given in ppm and were referenced to the resonance for residual solvent ($\delta$7.26 for CDCl$_3$ and 5.32 for CD$_2$Cl$_2$). Anhydrous CD$_2$Cl$_2$ (99.9 atom % D) was used for the NMR study of the hydrogen-bonding interactions between catalyst, monomer and propagating chain. A mixed deuterated solvent (CDCl$_3$/CF$_3$COOD = 2) was used for the $^1$H NMR characterization of polypeptide oligomer. Matrix-assisted laser desorption ionization time-of-flight (MALDI-TOF) mass spectra were collected on a Bruker UltraFLEX MALDI-TOF mass spectrometer in the reflector mode. 2,5-Dihydroxybenzoic acid (DHB) was used as matrices, and dichloromethane(85%)/trifluoroacetic acid (15%) was used as solvent. No cationic agent was added. Energy dispersive X-ray spectroscopy (EDS) analysis was carried out with a JSM-7500F scanning electron microscope (SEM, JEOL, Japan). The polymer film sample for EDS analysis was prepared by using solution spin coating method and was metalized with a sputtered gold layer before testing. The in situ FT-IR monitoring of NCA polymerization was carried out by using TENSOR II with MCT Detector from BRUKER. An attenuated total reflectance (ATR) diamond probe was connected to the reaction flask via AgX Fiber (Silver Halide, 9.5 mm × 1.5 m). Sampling is from 2800 cm$^{-1}$ to 650 cm$^{-1}$ at 8 wavenumber resolution and the

**Fig. 7** Proposed mechanism of ROP of Glu-NCA catalyzed by 1,3-Bis-HFAB. The nucleophilic ring-opening of the NCA by DMEA in the presence of 1,3-Bis-HFAB leads to the initiation of polymerization, whereby the ring-opened NCA releases one carbon dioxide molecule and forms the propagating amine for the subsequent addition of NCA monomers. During polymerization, multiple hydrogen-bonding equilibriums involving NCA monomer, initiator (DMEA), catalyst (1,3-Bis-HFAB), and propagating polymer chain exist. The catalysis proceeds by simultaneous activation of the carbonyl of NCA monomer and dynamic protection of the polymer chain-end amine group via hydrogen bonding to 1,3-Bis-HFAB

automatic sampling interval is 10 s. The real-time concentration of NCA was quantified by measuring the intensity of NCA's anhydride peak at $1792\ cm^{-1}$. Polymer characterizations were carried out by using Agilent 1260 infinity SEC instrument. The fractionation module was connected with three different detectors (from Wyatt Technology), Optilab T-rEX RI detector, ViscoStar-II viscometer and DAWN HELEOS-II multiangle laser-light scattering (MALLS) detector at a laser wavelength of 690 nm. Three $7.8 \times 300\ mm$ columns (Styragel® HT 2 DMF, Styragel® HT 3 DMF and Styragel® HT 4 DMF) and one guard column were used for polymer fractionation. The columns were calibrated with a series of polystyrene standards (Polymer Standard Service, U.S.A.). HPLC-grade DMF (containing 0.1 M LiBr) was used as the mobile phase at a flow rate of 1.0 mL/min. The whole SEC system was equilibrated in mobile phase at 60 °C. Polymer solutions with concentrations between 5.0 and 10.0 mg mL$^{-1}$ were injected with an injection volume of 200 μL. ASTRA software from Wyatt Technology was used for the data collection and analysis. The absolute molecular weights of resultant polypeptides were calculated with dn/dc values (dn/dc(PBLG) = 0.104 mL g$^{-1}$, dn/dc(PZLL) = 0.123 mL g$^{-1}$, dn/dc(PABLG) = 0.126 mL g$^{-1}$, dn/dc(PBLG-co-PZLL) = 0.114 mL g$^{-1}$, dn/dc(PBLG-co-PABLG) = 0.115 mL g$^{-1}$. PBLG, PZLL, and PABLG represent polypeptides obtained from ROP of Glu-NCA, Lys-NCA and Alkynyl-NCA, respectively.) The Density Functional Theory (DFT) calculation was performed at the GGA/PBE/DNP level of theory. All of the constructed structures are fully optimized and electrostatic potential (ESP)-fitting charges are derived from the DFT calculation.

**Polymerization procedure**. A typical procedure for the ring-opening polymerization of NCA was performed at 25 °C in a Braun Labmaster glovebox. Before polymerization, an attenuated total reflectance (ATR) diamond probe was connected with the reaction flask via a AgX Fiber (Silver Halide, 9.5 mm × 1.5 m). A solution of the initiator (DMEA) and catalyst (1,3-Bis-HFAB) in 4 mL of reaction solvent (DCM) was firstly added into the flask. After stirring at 25 °C for 1 min, the monomer solution of 0.4 g Glu-NCA monomer in 4 mL DCM was then quickly added into the flask to start the polymerization and the auto sampling system was switched on immediately. Sampling is from 2800 cm$^{-1}$ to 650 cm$^{-1}$ at 8

wavenumber resolution and the automatic sampling interval is 10 s. When full conversion was achieved from the in situ FT-IR spectra, 0.2 mL of the reaction mixture was taken out from the reaction system and diluted to 10 mg/mL by using HPLC-grade DMF (containing 0.1 M LiBr). The solution was then used for SEC characterization to determine the molecular weight and molecular weight distribution of the obtained polypeptides. The remaining reaction mixture was poured into methanol, sonicated, and centrifuged to remove the solvent. The white precipitate was collected and further washed twice with methanol and then dried overnight under vacuum to give the final polymer. Polypeptide with variable molecular weights was prepared by tuning the relative ratios of monomer to initiator [NCA]/[DMEA]. Homopolymerization of Lys-NCA or Alkynyl-NCA was operated by following above operations at 25 °C with [NCA]$_0$ = 0.19 M. As for random copolymerizations of Glu-NCA with Lys-NCA or Alkynyl-NCA, the overall monomer concentration [Glu-NCA + Lys-NCA]$_0$ or [Glu-NCA + Alkynyl-NCA]$_0$ was fixed at 0.19 M. As for the sequential polymerization of Glu-NCA with Lys-NCA or Alkynyl-NCA, the polymerization of Glu-NCA was carried out firstly. When a full conversion of the Glu-NCA was achieved, the Lys-NCA or Alkynyl-NCA monomer was quickly added into reaction system. The whole polymerization process was monitored by in situ FT-IR in order to determine the adding time of the second kind of monomer.

**Cell experiments**. The human liver cancer cell line HepG2 was purchased from ATCC (American Type Culture Collection, VA). Before experiments, the cell was kept at 37 °C in a humidified atmosphere of 5% CO$_2$ in dulbecco's modified eagle medium (with 10% fetal bovine serum, 25 mmol L$^{-1}$ hydroxyethyl piperazine ethanesulfonic acid buffer, 100 U mL$^{-1}$ penicillin, and 100 μg mL$^{-1}$ streptomycin). The extract of the polypeptide sample was prepared by following international standard ISO 10993-12:2012. The cellular toxicity of the polypeptide samples was evaluated as follows. HepG2 cells were firstly seeded in 96-well plates with a density of $5 \times 10^3$ cells per well and incubated overnight. Subsequently, the cells were incubated for 48 h with various concentrations of the extract of polypeptides. Afterwards, the cytotoxicity was evaluated by adding Cell Counting Kit-8 (CCK-8) solution (from RayBiotech, Inc.) to each well of the plate. After incubation for 1 h,

the absorbance was investigated at 450 nm using a Multiskan MK3 Microplate Reader. For fluorescence microscopy, a live-dead cell staining kit from RayBiotech, Inc. was used. The live-dead cell staining kit provides the ready-to-use reagents for convenient discrimination between live and dead cells. The kit utilizes Live-Dye$^{TM}$, a cell-permeable green fluorescent dye (Ex/Em = 488/518 nm), to stain live cells. Dead cells can be easily stained by propidium iodide (PI), a cell non-permeable red fluorescent dye (Ex/Em = 488/615). Stained live and dead cells can be visualized by fluorescence microscopy using a band-pass filter.

## Data availability

The source data for Figs. 3–6 and Supplementary Figs, 2–10, 12–22 and 24–26 are provided with this paper. These source data were also deposited in the Figshare (https://doi.org/10.6084/m9.figshare.8855750).

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

## Acknowledgements
This work is supported by grants from the National Natural Science Foundation of China (No. 21774071) and Natural Science Foundation of Shaanxi University of Science and Technology (No. 2017QNBJ-07). W.Z. thanks the support from the Youth Hundred-Talent Program of Shaanxi Province (No.SXBR9227), the National High-Level Foreign Expert Project (No.GDT20186100425), Biomass Chemistry and Materials Academician Workstation Project in SUST (No.134090002) and Northwest Polytechnic University High Performance Computing and Development Center. N.H. acknowledges the support of King Abdullah University of Science and Technology.

## Author contributions
W.Z., Y.N., and N.H. conceived the idea, designed the experiments, evaluated the data, and wrote the manuscript together. Y.L. and Z.F performed majority of the experiments and summarized results. J.L. carried out the computational studies. All of the authors reviewed, approved, and contributed to the final version of the manuscript.

## Additional information

**Competing interests:** The authors declare no competing interests.

