## [Peer Review File · Nature Communications]

Reviewers' comments:

Reviewer #1 (Remarks to the Author):

This paper reports fluorinate alcohol-induced rapid and living ring-opening polymerization of N-carboxylic anhydride (NCA) for the synthesis of well-defined polypeptide without using any metal catalysts. The results are interesting and show that the fluorinated diol (1,3-bis-HFAB: 1,3-bis(2-hydroxyhexafluoroisopropyl)benzene)) efficiently works for the rapid living polymerization of NCA to enable the synthesis of high molecular weight polypeptide. The interactions between 1,3-bis-HFAB and monomer, initiator, or chain-end were well studied using ^1H and ^{19}F NMR and computer simulation. However, the following points should be investigated. Without any major revisions, it would not be suitable for publication in Nature Communications.

1. Introduction. This paper reports ring-opening polymerization of NCA, but almost no backgrounds about the former works on NCA are written. The introduction only focuses on just a conceptual matter and lacks scientific depth about the polymerization of NCA. I do not fully understand what is new in comparison to the reported works on NCA and what is a problem in this field. A more detail about ring-opening polymerization of NCA should be described in Introduction by emphasizing what is new as ROP of NCA.
2. In polymerization of NCA, usually a primary amine, which has a similar structure to the propagating terminal, may be used and more suitable. Alcohol may not be a good initiator in terms of its low nucleophilicity. I do not understand why the aminoalcohol (DMEA) is used as an initiator. The unsymmetrical SEC curves with a slight tailing in the lower molecular weight region (Figure 1) suggests slow initiation from DMEA.
3. Scheme 2. I do not understand meaning of "overactive" species of primary amine terminals. This word is just a concept without any proofs. To confirm that they are overactive, additional experiments using a primary amine as an initiator in the absence of 1,3-bis-HFAB are necessary.
4. In addition, to show that another polymer terminal with tertiary amine structure is "harmful", the control experiments with a tertiary amine as a potential initiator in the absence of 1,3-bis-HFAB is also necessary.
5. The term "shuttling" is merely a catchy word and does not express the working mechanism of 1,3-bis-HFAB properly. The word "shuttle" originally means what travels frequently between two places and what transports people or something between two places. Indeed, a shuttle bus moves between two places, and a shuttle in badminton moves between two players. In addition, chain-shuttling agents, which have already been used in coordination polymerization of olefins, transport polymer chains between two catalysts. However, in this paper, 1,3-bis-HFAB moves among many polymer chains and transports nothing. Such an inappropriate usage may cause fatal misunderstanding. In general, the introduction seems too superficial and just plays with catchy words.
6. TOF and TON used in this paper are meaningless. These values are just used to show how many monomers react with propagating chain ends per chain. They are very strange for many polymer chemists working on living polymerizations. Usually, TOF and TON are used for catalysts. The propagating chain ends in this case are not catalysts. The real catalyst may be 1,3-bis-HFAB. I would not recommend the use of TOF and TON in this paper.
7. According to the plots in Figure 4, the first-order dependences were only observed over 50% conversion of monomer ($\ln([M]_0/[M]_t) > 0.7$). What happened below 50% conversion of monomer? Therefore, the expression, "A first-order dependence was observed in each case", must be wrong. A more detailed kinetic analysis is necessary. Otherwise, the value for the kinetic order in catalyst, -0.40, is meaningless and may be wrong. I do not understand what mechanism the kinetic order -0.40 means.
8. Again, I would strongly recommend that the authors investigate the polymerization of NCA using silyl-protected ethanolamine (DMES-TMS') without 1,3-bis-HFAB to show how "harmful" the tertiary amine is for the polymerization.
9. Lines 216-220. I would not agree. If the activated monomer mechanism is also involved in the polymerization, the molecular weights should be lower than the theoretical one because additional propagating chains form in addition to the initiator.
10. The following papers may be cited. Nat. Rev. Chem. 1, 88 (2017) for a review on hexafluoroisopropanol (HFIP), which is a more common fluorinated-alcohol in organic reactions; Macromolecules 39, 4054-4061 (2006) and Chem. Rev. 109, 5120-5156 (2009) for the use of 1,3-bis-HFAB for stereospecific radical polymerization using hydrogen-bonding interaction.

11. It would be also nice to use HFIP for the polymerization in place of the diol (1,3-bis-HFAB). The results would clarify the effective structure of fluorinated alcohol and the polymerization mechanism.

Reviewer #2 (Remarks to the Author):

The manuscript of Zhao et al. reports on the organocatalytic ROP of Glu-NCA by the 1,3-bis-HFAB catalyst in presence of DMEA as initiator. In contradiction to what authors claimed, such a manuscript does not provide a green approach to access well-defined polypeptides. Only one type of polypeptide was prepared, and the "green" term is used erroneously. If it is correct to claim that fluorinated alcohols have attracted extensive research interest due to their favorable properties, the 1,3-bis-HFAB catalyst used in that study is highly irritant and corrosive. To what it has been reported and to claim that such polymers could be of use for biomedical applications, it would be of interest to verify that all the catalyst is removed during/after the polymer workup and, in case of a "stuck" catalyst, the as-prepared polypeptide does not show any sign of toxicity. However, and very interestingly, the high-rate of reaction and the high-selectivity in terms of mechanisms (no AMM side reaction reported) are outstandingly interesting and represent a significant advance in the field.

In my opinion this work is well presented and deserves publication but not as a Nature Communications in its present form.

Reviewer #3 (Remarks to the Author):

This paper report the controlled or living ring opening polymerization of NCA using aminoalcohol (DMEA) as the initiator, and a small molecular fluorinated alcohol, 1,3-bis-HFAB, as an organocatalyst. Through analysis of molecular structure of the polymer product and measuring polymerization kinetics, the authors proved that the ring opening polymerization was living and the chain growth was mediated by the reversible activation-deactivation of the initiating hydroxyl group and the propagating terminal amine group. A thorough study on NMR spectroscopy of the model compounds indicated that the activation-deactivation process was fulfilled through labile hydrogen bonding between HFAB and the corresponding species. In my opinion, the results in the present manuscript represent a remarkable progress in ring opening polymerization of NCA because the initiating/catalytic system is all organic and the polymerization is of high selectivity. On this basis, I recommend publication of the work by Nature Communication. The article was well written and the conclusion was logically supported by the results. I only find a few typos, as follows, which need minor revision before publication,

- 1) Line 100: Why you say the experimental M_n ,SEC was calculated based on NMR? As far as I understand, the molecular weight should be M_n ,SEC-LS.
- 2) Line 140 & 216: Here should be MALDI-TOF MS spectrum.
- 3) Line 169: The dependence of what? The sentence must be revised to be complete.

Response to the referees' comments:

Reviewer #1 (Remarks to the Author):

This paper reports fluorinate alcohol-induced rapid and living ring-opening polymerization of N-carboxylic anhydride (NCA) for the synthesis of well-defined polypeptide without using any metal catalysts. The results are interesting and show that the fluorinated diol (1,3-bis-HFAB: 1,3-bis(2-hydroxyhexafluoroisopropylbenzene)) efficiently works for the rapid living polymerization of NCA to enable the synthesis of high molecular weight polypeptide. The interactions between 1,3-bis-HFAB and monomer, initiator, or chain-end were well studied using ¹H and ¹⁹F NMR and computer simulation. However, the following points should be investigated. Without any major revisions, it would not be suitable for publication in Nature Communications.

1. Introduction. This paper reports ring-opening polymerization of NCA, but almost no backgrounds about the former works on NCA are written. The introduction only focuses on just a conceptual matter and lacks scientific depth about the polymerization of NCA. I do not fully understand what is new in comparison to the reported works on NCA and what is a problem in this field. A more detail about ring-opening polymerization of NCA should be described in Introduction by emphasizing what is new as ROP of NCA.

Response:

We sincerely thank the reviewer for these valuable suggestions to help us improve the manuscript. In the revised Introduction, we have discussed the issues/challenges and the current research status of NCA polymerization. Furthermore, we have compared/distinguished our work with/from those in the literature, as detailed below.

The chemistry of NCAs is unique: the carbonyl group (C5) of NCA is highly electrophilic and can be readily attacked by various nucleophiles, inducing ROP (*via* the “normal amine mechanism”, Path 1 in Scheme S1). Furthermore, the acidic proton at the N3 position can be readily deprotonated under basic conditions. The resultant NCA nucleophilic anion can attack the carbonyl group (C5) of another NCA molecule, triggering an uncontrolled ROP of NCA, the so-called “activated monomer mechanism” (AMM, Path 2 in Scheme S1), or can undergo rearrangement to form

isocyanocarboxylates, which terminate the growing chains.

The “normal amine” (NAM) and “activated monomer” (AMM) mechanisms, often compete each other and complicate the overall polymerization process, making it challenging to produce well-defined (co)polypeptides (*Chem. Rev.* 2009, 109, 5528; *Adv. Mater.* 1997, 9, 299; *Adv. Polym. Sci.* 2006, 202, 1.). In 1997, Deming reported the first example of controlled/living ROP of NCAs by using transition metal complexes instead of the traditional primary amine as active species to control the monomer addition at the polymer chain-end (T. J. Deming, *Nature*, 1997, 390, 386). Since then, a few studies on controlled NCA polymerizations have been reported, exploring novel initiators (e.g. Pt complexes, ammonium salts, silazane derivatives, rare earth metal complexes, amine-borane Lewis pairs and Li complex) or new conditions for the traditional primary amine-initiated ROP (e.g. high vacuum, low temperature and nitrogen flowing). (*Chem. Commun.*, 2003, 2944; *J. Am. Chem. Soc.*, 2007, 129, 14114; *Macromolecules*, 2008, 41, 3455; *Biomacromolecules*, 2004 5, 1653.). As for the ROP of NCA promoted by metal complexes, the metal residues in resultant polypeptides can cause toxicity concerns. An extra purification step of the as-synthesized polypeptide, such as dialysis is usually required. In regards to the NCA polymerizations initiated by organic initiators, low polymerization activities are usually observed, even with low monomer/initiator ratios (*Nature Communications*, 9, doi:10.1038/s41467-018-07711-y), showing limitations in the synthesis of polypeptides with high molecular weights and/or for ROP of NCAs with low stabilities. Therefore, developing a fast and metal-free process for the synthesis of polypeptides is very necessary and promising.

Scheme S1 Competition of the “normal amine mechanism” (NAM, Path 1) and “activated monomer mechanism” (AMM, Path 2) in the ROP of α -amino acid NCAs.

2. In polymerization of NCA, usually a primary amine, which has a similar structure to the propagating terminal, may be used and more suitable. Alcohol may not be a good initiator in terms of its low nucleophilicity. I do not understand why the aminoalcohol (DMEA) is used as an initiator. The unsymmetrical SEC curves with a slight tailing in the lower molecular weight region (Figure 1) suggests slow initiation from DMEA.

Response:

We thank the reviewer for the comments. We agree with the reviewer’s point that “the primary amine has a similar structure to the propagating terminal, may be used and more suitable”. Unfortunately, the primary amine can act as both a nucleophile and a base. When primary amine is used to initiate NCA polymerization, it can function as a nucleophile to attack the carbonyl group (C5) of NCA to trigger the polymerization via the “normal amine mechanism” (NAM); at the same time the primary amine is basic enough to deprotonate the acidic proton at the N3 position of NCA, forming NCA anion, which can lead to unwanted polymerizations in the system via the “activated monomer mechanism” (AMM).

According to a previous study, after the polymerization initiation, the primary amine at the end of the polymer chain is also prone to side-reactions with solvents, end-group termination, and competition with the desirable polymerization. As a result, during the polymerization process, the propagating reactions will switch back and forth between the NAM and AMM mechanisms, complicating the polymerization process, and thus making the synthesis of well-defined polypeptides challenging. (T. J. Deming, *Adv. Mater.*, 1997, 9, 299–311).

Yes, the AMM pathway can be largely suppressed by lowering the reaction temperature; however, a significant decrease of polymerization activity is usually observed and multi-day reaction time is needed to complete the polymerization, even with a low monomer/initiator ratio (*Polym. Chem.*, 2011, 2, 1322), which is adverse to NCA polymerization, especially for NCA monomers with low stability.

For the reasons stated above, the present strategy of using aminialcohol/**1,3-bis-HFAB** catalytic system, to promote a fast NCA polymerization is highly attractive and innovative.

The aminoalcohol (**DMEA**) used in this study is different from common alcohols. The nitrogen atom at the beta-position (vs. hydroxyl group) in **DMEA**, can form

intramolecular hydrogen-bonding with the hydroxyl group, which is supported by the computer simulation in our original MS (Figure S25). This interaction can increase the nucleophilicity of the hydroxyl group and facilitate the nucleophilic attack of hydroxyl group on the carbonyl group (C5) of NCA monomer, resulting in a fast initiation of the NCA polymerization in the presence of fluorinated catalyst. Faster initiation than chain propagation is essential for obtaining polymers with low polydispersity. Furthermore, the hydroxyl group is not basic (in contrast to primary amine), thus preventing the deprotonation of N3 of the NCA monomer at the initiating step and minimize the polymerization triggered by NCA anion via the AMM pathway. Another advantage of using **DMEA** is that the nitrogen atom of **DMEA** can provide a heteroatom reactive site for the post-functionalization of resultant polypeptides, which cannot be achieved by common primary amines.

In regards to the SEC traces, we carefully studied the reported catalytic/initiator systems for controlled/living NCA polymerization in the literature and found that the SEC traces of the polypeptides made are also not symmetrical and have slight tailings in a low molecular weight region (*Nature*, 1997, 390, 386; *Chem. Commun.*, 2003, 2944; *J. Am. Chem. Soc.*, 2007, 129, 14114; *Macromolecules*, 2008, 41, 3455; *J. Polym. Sci., Part A: Polym. Chem.*, 2012, 50, 1076). Two examples are given below.

We further checked the SEC traces of very common polymers such as PS and PMMA prepared by controlled/living radical polymerization via ATRP, RAFT, and NMP and found that the results are similar (Two examples are given below). The explanations are as follows:

(1) The molecular weight distribution for ideal living polymerization usually follows a Poisson distribution, which is not a symmetric distribution. A Gaussian distribution (symmetric) approximation for the Poisson distribution can only be used for a very large number of events (*Nuclear Instruments and Methods in Physics Research*, 1984, 228, 120).

(2) The inevitable instrumental issue of SEC, namely the “band broadening effect”, can also increase the asymmetry of the molecular weight distribution of the measured polymer, especially when the molecular weight of the polymer is high (*Macromol. Theory Simul.* 2011, 20, 667; *J. Liq. Chromatogr. Relat. Technol.* 2002, 25, 1962.).

On the other hand, we cannot rule out the possibility of the presence of minor side reactions in the system, (although the D of polypeptide prepared in our study is very low, below 1.08). The novel **1,3-bis-HFAB/DMEA** system developed for NCA polymerization in this study relies on multi-dynamic H-bonding interactions between catalyst, monomers and chain-ends, the side reactions can be greatly suppressed, but in theory, it cannot be completely eliminated. This is similar to the controlled/living radical polymerization, regardless of the great progress made, the complete

elimination of side reactions (biradical termination reactions) would be practically impossible in theory.

Fig. 1 Mass distributions (SEC, left) and sedimentation coefficient distributions (AUC, right) of the PS-PZLLys samples 2 (dashed line) and 6 (solid line).

The SEC traces (left) of polystyrene-*b*-polypeptide

From: Chem. Commun., 2003, 2944

Fig. 2 HFIP SEC results of BLG polymerization (DP: 100): (a) at 20 °C under nitrogen and (b) at 20 °C under high vacuum.

The SEC traces of polypeptide made under nitrogen (a) and high vacuum (b)

From: Polym. Chem., 2011, 2, 1322–1330

SEC traces of different polystyrenes synthesized by ATRP.

From: ACS Symposium Series, 2009,1023,343-355

SEC traces of different poly(methyl methacrylate)s synthesized by RAFT

From: Polymer 2005, 46, 8458

3. Scheme 2. I do not understand meaning of “overactive” species of primary amine terminals. This word is just a concept without any proofs. To confirm that they are overactive, additional experiments using a primary amine as an initiator in the absence of 1,3-bis-HFAB are necessary.

Response:

Previous studies on NCA polymerization revealed that the primary amine chain-end group is prone to side-reactions (TraV. Sluyterman, A.; Labruyere, A. *Recl. Chim.* **1954**, 73, 347; Diekmann, W.; Breest, F. *Ber.* **1906**, 39, 3052; *Polym. Chem.* **2011**, 2, 1322; Kricheldorf, H. R. *α -Aminoacid-N-Carboxyanhydrides and Related Materials*, Springer, Berlin Heidelberg New York, 1987; Kricheldorf, H. R. In: Penczek S (ed) *Models of Biopolymers by Ring-Opening Polymerization*. CRC Press, Boca Raton, FL, 1990; Deming, T. J., *Adv. Mater.*, **1997**, 9, 299.). This is why researchers have tried to explore new initiators such as transition metal complexes (*Nature*, **1997**, 390, 386), silazane derivatives (*J. Am. Chem. Soc.*, **2007**, 129, 14114.), Pt-based and rare earth metal complexes (*Macromolecules*, **2008**, 41, 3455; *J. Polym. Sci., Part A: Polym. Chem.*, **2012**, 50, 1076) to replace the primary amine terminals with new active species, aiming to achieve controlled/living ROP of NCA.

We followed your suggestion and carried out the ROP of NCA initiated by a primary amine, n-hexylamine (**HA**), in the absence of **1,3-bis-HFAB** (polymerization conditions: [Glu-NCA]/[n-hexylamine]=120, [Glu-NCA]=0.19M, 25°C). The results showed that the polymerization was very slow, forming polypeptide with a broad molecular weight distribution (4.5h, 98% conv., $M_{n,SEC-MALS}=3.23 \times 10^4$, $\bar{D}=1.17$,

$M_{n,calcd}=2.57\times 10^4$, Figure S9). A further kinetics study via *in situ* FT-IR method showed that the polymerization process did not have a constant reaction rate. When the monomer conversion was above 82%, the polymerization slowed down and deviated from the living polymerization due to side reactions (Figure S9). When the monomer/initiator ratio was increased from 120 to 240, the results were even worse. Only 68% of monomer was converted to polymer in 9 hours. The resultant polymer showed a bimodal SEC trace (with a shoulder peak) and a higher molecular weight ($M_{n,SEC-MALS}=6.10\times 10^4$) than the theoretical one ($M_{n,calcd}=3.58\times 10^4$) (Figure S9). Also, the monomer could not be completely converted to polypeptides even in long reaction time (one day). These results and their discussion are included in the revised manuscript.

Figure S9 Kinetics study of ROP of NCA initiated by n-hexylamine and the corresponding SEC traces of resultant polypeptide (polymerization conditions: [Glu-NCA]/[n-hexylamine]=120 and 240, [Glu-NCA]=0.19M, 25°C, CH₂Cl₂). When the monomer/initiator ratio was increased from 120 to 240, a shoulder was observed in the SEC trace of resultant polypeptide and the $M_{n,SEC-MALS}$ is nearly two times that of $M_{n,calcd}$, indicating poor control of polymerization.

4. In addition, to show that another polymer terminal with tertiary amine structure is “harmful”, the control experiments with a tertiary amine as a potential initiator in the absence of 1,3-bis-HFAB is also necessary.

Response:

We followed your suggestions, and carried out the control experiment by using tertiary amine, triethylamine (TEA), as a potential initiator in the absence of 1,3-bis-HFAB (conditions: [Glu-NCA]/[TEA]=120, [NCA]₀=0.19M, 25°C, CH₂Cl₂). The results showed that the polymerizations were very slow (24h, 90% conv.) and uncontrollable, resulting in polypeptides with very broad unsymmetrical SEC trace (*D*=1.80) and much higher molecular weight ($M_{n,SEC-MALS}=3.92\times 10^5$) than theoretical one ($M_{n,calcd}=2.37\times 10^4$) (Figure S24). The $M_{n,SEC-MALS}$ is 16 times higher than that of the calculated molecular weight. The kinetics study showed that a first-order dependence of polymerization rate on monomer concentration was not observed, indicating slow initiating of the polymerization.

Figure S24 (a) Kinetics study of ROP of Glu-NCA initiated by TEA in the absence of 1,3-bis-HFAB, conditions: [Glu-NCA]/[TEA]=120, [NCA]₀=0.19M, 25°C, CH₂Cl₂; (b) corresponding SEC traces of obtained polypeptide ($M_{n,SEC-MALS}=3.92\times 10^5$, *D*=1.80, the $M_{n,SEC-MALS}$ is 16 times that of the calculated molecular weight ($M_{n,calcd}=2.37\times 10^4$), indicating a slow initiating of the polymerization.)

5. The term “shuttling” is merely a catchy word and does not express the working mechanism of 1,3-bis-HFAB properly. The word “shuttle” originally means what travels frequently between two places and what transports people or something between two places. Indeed, a shuttle bus moves between two places, and a shuttle in badminton moves between two players. In addition, chain-shuttling agents, which have already been used in coordination polymerization of olefins, transport polymer chains between two catalysts. However, in this paper, 1,3-bis-HFAB moves among

many polymer chains and transports nothing. Such an inappropriate usage may cause fatal misunderstanding. In general, the introduction seems too superficial and just plays with catchy words.

Response:

We thank the reviewer and have changed the word “shuttling” to “moving” to avoid the misunderstanding. In addition, we have carefully revised the introduction, added the current research status of NCA polymerization, and compared our work with those reported in the literature.

6. TOF and TON used in this paper are meaningless. These values are just used to show how many monomers react with propagating chain ends per chain. They are very strange for many polymer chemists working on living polymerizations. Usually, TOF and TON are used for catalysts. The propagating chain ends in this case are not catalysts. The real catalyst may be 1,3-bis-HFAB. I would not recommend the use of TOF and TON in this paper.

Response:

Thank you for your comments.

We followed your suggestion and deleted “TON” and “TOF” in the revised MS.

7. According to the plots in Figure 4, the first-order dependences were only observed over 50% conversion of monomer ($\ln([M]_0/[M]_t) > 0.7$). What happened below 50% conversion of monomer? Therefore, the expression, “A first-order dependence was observed in each case”, must be wrong. A more detailed kinetic analysis is necessary. Otherwise, the value for the kinetic order in catalyst, -0.40, is meaningless and may be wrong. I do not understand what mechanism the kinetic order -0.40 means.

Response:

According to previous kinetics studies, ROP of Glu-NCA in several solvents showed two successive rates following a relatively rapid initiation (*J. Am. Chem. Soc.* 1957, 79, 3961; *J. Am. Chem. Soc.* 1957, 79, 3948.). The rate constant for the second propagation step is at least several times higher than that for the first step. The whole polymerization rate is largely dependent on the second stage propagation. The

two-stage propagation is due to the transition of polypeptide chain from β to α configuration during growth. The β -polypeptide (non-helical, low DP), is responsible for the slow reaction and the α -polypeptide (helical, high DP) for the fast reaction. When the polymerization proceeds from the slow phase to the fast phase, a sharp turning point may not be easily identifiable, especially when the polymerization rate is too fast.

We repeated the kinetics study by using a high monomer/initiator ratio of 480 and varying the catalyst concentration from 1.7% to 17 mol% (conditions: $[\text{NCA}]/[\text{DMEA}]=480$, $[\mathbf{1,3\text{-Bis-HFAB}}]_0= 1.7, 3.3, 5.0, 6.7$ and 17mol\% vs. $[\text{NCA}]_0$, $[\text{NCA}]_0=0.19\text{M}$, 25°C , CH_2Cl_2). Such conditions would slow down the polymerization, which makes it possible to get more details. The first-order dependence can be observed from 10% to 95% conversion of monomer ($\ln([\text{M}]_0/[\text{M}]_t)$ values are from 0.1 to 3) (Figure 4 in revised MS). The value of the kinetic order is -0.3785 which is slightly different from the previous value (-0.3930) presented in the original MS, but still close to -0.40 . This result indicates that excessive loading of catalysts in the system can slow down the polymerization, consistent with that of the *in situ* monitoring of the polymerization rate by adding excess catalyst in the system during polymerization (Figure 8 in revised MS).

When the excess catalyst is used, the chemical equilibrium between the active terminal amine and the dormant terminal amine will shift to the side of dormant terminal amine and result in more dormant species in the system, which will slow down the polymerization.

Figure 4 Kinetics study of Glu-NCA polymerization promoted by $\mathbf{1,3\text{-Bis-HFAB}}$ at various catalyst concentrations ($[\mathbf{1,3\text{-Bis-HFAB}}]_0=1.7\text{-}17\text{mol\%}$ vs. $[\text{NCA}]_0$) and fixed $[\text{DMEA}]_0$ (4.0×10^{-4} M) and $[\text{NCA}]_0$ (0.19 M) concentrations. Polymerizations carried out in DCM at room temperature

8. Again, I would strongly recommend that the authors investigate the polymerization of NCA using silyl-protected ethanolamine (DMEA-TMS') without 1,3-bis-HFAB to show how "harmful" the tertiary amine is for the polymerization.

Response:

We took the suggestion from the reviewer and supplemented the control experiments by using silyl-protected ethanolamine (**DMEA-TMS'**) for NCA polymerization in the absence of **1,3-bis-HFAB**. The results showed that the polymerization was very slow (24h, 92% monomer conversion) and uncontrolled, giving polypeptide with a much higher molecular weight than the theoretical one and broad molecular weight distribution ($M_{n,SEC-MALS}=2.67\times 10^5$, $D=1.63$, Figure S5). The $M_{n,SEC-MALS}$ is ten times that of the calculated molecular weight ($M_{n,calcd}=2.42\times 10^4$). Kinetics studies show that a first-order dependence of polymerization rate on monomer concentration was not observed, indicating a slow initiation of the polymerization. In the absence of **1,3-bis-HFAB**, the polymerization of NCA is triggered by the tertiary amine of **DMEA-TMS'** via the "activated monomer mechanism". We have added these results in SI.

Figure S5 (a) Kinetics study of ROP of Glu-NCA initiated by **DMEA-TMS'** in the absence of **1,3-bis-HFAB**, conditions: $[Glu-NCA]/[DMEA-TMS']=120$, $[NCA]_0=0.19M$, $25^\circ C$, CH_2Cl_2 ; (b) corresponding SEC traces of obtained polypeptide ($M_{n,SEC-MALS}=2.67\times 10^5$, $D=1.63$, the $M_{n,SEC-MALS}$ is ten times that of the calculated molecular weight ($M_{n,calcd}=2.42\times 10^4$), indicating the polymerization proceeds through activated monomer mechanism)

9. Lines 216-220. I would not agree. If the activated monomer mechanism is also involved in the polymerization, the molecular weights should be lower than the

theoretical one because additional propagating chains form in addition to the initiator.

Response:

Thank you for your comments.

In this paper, we have presented an innovative strategy that leads to high-rate and high-selectivity controlled/living polymerizations of NCA (essentially eliminating the AMM-induced side reactions).

From the literature, if the “activated monomer mechanism” is involved in the polymerization, NCA anions must have been generated in the system. The NCA anions can rearrange to α -isocyanatocarboxylates, which can react with the amino chain-end and terminate the polymer chain (see below scheme (i)). (K. D. Kopple, *J. Am. Chem Soc.* 1957, 79, 6442; Diekmann, W.; Breest, *F. Ber.* 1906, 39, 3052). The non-reversible reactions between α -isocyanatocarboxylates and amino chain-end can form more α -isocyanatocarboxylates, and terminate more polymer chains (Kricheldorf H. R., *α -Aminoacid-N-Carboxyanhydrides and Related Materials*. Springer, Berlin Heidelberg New York, 1987). Also, if NCA anions are formed during the “amine” chain growth polymerization, a chain transfer process can take place, which terminates the chain growth and simultaneously generate NCA anions (see below scheme (ii)).

Therefore, if the “activated monomer mechanism” is involved in the polymerization, the number of propagating chains in the system will decrease, increasing the molecular weight of the resultant polypeptide. At the same time, the D of obtained polypeptides will also increase (as a result of two active species with different activities, primary amine, and NCA anion).

To reinforce the above point, we performed additional experiments by studying the ROP of NCA initiated by *N,N*-dimethyl-1,2-ethanediamine (**DMEDA**, $(\text{CH}_3)_2\text{NCH}_2\text{NH}_2$) in the absence of **1,3-bis-HFAB** (polymerization conditions: $[\text{Glu-NCA}]/[\text{DMEDA}]=100$, $[\text{Glu-NCA}]=0.19\text{M}$, 25°C). **DMEDA** has both tertiary amine and primary amine in its molecular structure and can trigger the NCA polymerization via both mechanisms. The SEC characterization demonstrated that the resultant polypeptide possessed a bimodal molecular weight distribution ($D=1.38$) and the experimental molecular weight ($M_{n,\text{SEC-MALS}}=4.71\times 10^4$) is more than two times higher than the theoretical molecular weight ($M_{n,\text{calcd}}=2.19\times 10^4$) (Figure S6).

Figure S6 SEC traces of polypeptide prepared by ROP of Glu-NCA initiated by **DMEDA** (conditions: $[\text{Glu-NCA}]/[\text{DMEDA}]=100$, $[\text{Glu-NCA}]=0.19\text{M}$, 25°C)

10. The following papers may be cited. *Nat. Rev. Chem.* 1, 88 (2017) for a review on hexafluoroisopropanol (HFIP), which is a more common fluorinated-alcohol in organic reactions; *Macromolecules* 39, 4054-4061 (2006) and *Chem. Rev.* 109, 5120-5156 (2009) for the use of 1,3-bis-HFAB for stereospecific radical polymerization using hydrogen-bonding interaction.

Response:

Thank you for your suggestions. We have cited these references in the revised MS.

11. It would be also nice to use HFIP for the polymerization in place of the diol (1,3-bis-HFAB). The results would clarify the effective structure of fluorinated alcohol and the polymerization mechanism.

Response:

Thank you for your suggestions.

We followed your suggestion and carried out experiments using **HFIP** instead of **1,3-bis-HFAB** (polymerization conditions: $[\text{Glu-NCA}]/[\text{DMEA}]/[\text{HFIP}]=120/1/1$, $[\text{Glu-NCA}]=0.19\text{M}$, 25°C).

As expected, the results are not as good as those using **1,3-bis-HFAB**. The resultant polypeptide has a bimodal molecular weight distribution, indicating that **HFIP** may participate in the initiation (presumably because of lower steric hindrance) and/or has no capability to silent the initiating action of tertiary amine in **DMEA**. We monitored the NCA polymerization with **HFIP** by using *in situ* FT-IR (polymerization conditions: $[\text{Glu-NCA}]/[\text{HFIP}]=60$, $[\text{Glu-NCA}]=0.19\text{M}$, 25°C) and found that **HFIP** initiated the NCA polymerization in a slow and uncontrolled manner (4.4h, 85% conversion, $M_{n,\text{SEC-MALS}}=3.44\times 10^4$, $D=1.22$, Figure S22). We also studied the NCA polymerization by using silyl-protected ethanolamine (**DMEA-TMS'**)/**HFIP** (polymerization conditions: $[\text{Glu-NCA}]/[\text{DMEA-TMS}']/[\text{HFIP}]=120/1/1$, $[\text{Glu-NCA}]=0.19\text{M}$, 25°C). The polymerization is slow (3h, 93% conversion) and the resultant polypeptide has a much higher molecular weight than the theoretical one and a bimodal molecular weight distribution ($M_{n,\text{SEC-MALS}}=2.67\times 10^5$, $D=1.63$, Figure S23). This is because **HFIP** participates in the initiation and cannot silent the tertiary amine in **DMEA**, so both the NAM and AMM pathways co-exist in the polymerization system.

In contrast, for **1,3-bis-HFAB** used in this study, the bulky electron-withdrawing fluorinated groups can increase the acidity of the alcohol (increasing hydrogen bonding), while steric factors reduce the nucleophilicity of the alcohols and prevent its participation in initiation. That is why **1,3-bis-HFAB** shows much better results than **HFIP**.

We have added the above results in the supporting materials of the revised MS.

Figure S22 Kinetics study of ROP of Glu-NCA in the presence of **HFIP** and corresponding SEC traces of obtained polypeptide ($M_{n,SEC-MALS}=3.44\times 10^4$, $D=1.22$); conditions: $[\text{Glu-NCA}]/[\text{HFIP}]=60$, $[\text{NCA}]_0=0.19\text{M}$, 25°C , CH_2Cl_2 . A first-order dependence of polymerization rate on the monomer concentration was not observed in kinetics study, indicating a slow initiation of the polymerization.

Figure S23 Kinetics study of ROP of Glu-NCA initiated by **DMES-TMS'**/HFIP and corresponding SEC traces of obtained polypeptide ($M_{n,SEC-MALS}=2.67\times 10^5$, $D=1.63$); conditions: $[\text{Glu-NCA}]/[\text{DMES-TMS}']/[\text{HFIP}]=120/1/1$, $[\text{NCA}]_0=0.19\text{M}$, 25°C , CH_2Cl_2 ; A first-order dependence of polymerization rate on the monomer concentration was not observed in kinetics study, indicating a slow initiation of the polymerization

Reviewer #2

The manuscript of Zhao et al. reports on the organocatalytic ROP of Glu-NCA by the 1,3-bis-HFAB catalyst in presence of DMEA as initiator. In contradiction to what authors claimed, such a manuscript does not provide a green approach to access well-defined polypeptides. Only one type of polypeptide was prepared, and the “green” term is used erroneously. If it is correct to claim that fluorinated alcohols have attracted extensive research interest due to their favorable properties, the 1,3-bis-HFAB catalyst used in that study is highly irritant and corrosive. To what it has been reported and to claim that such polymers could be of use for biomedical applications, it would be of interest to verify that all the catalyst is removed during/after the polymer workup and, in case of a “stuck” catalyst, the as-prepared polypeptide does not show any sign of toxicity.

However, and very interestingly, the high-rate of reaction and the high-selectivity in terms of mechanisms (no AMM side reaction reported) are outstandingly interesting and represent a significant advance in the field.

In my opinion this work is well presented and deserves publication but not as a Nature Communications in its present form.

Response:

We thank the reviewer for these valuable suggestions to help us improve our manuscript. In order to explore the generality of this new strategy for polypeptide synthesis, we have supplemented the homo-/co-polymerization of several different NCA monomers (such as Lys-NCA and Alkynyl-NCA) and prepared a series of well-defined (co)polypeptides by using this new strategy. These data were included in the new Table 1 of the revised manuscript (runs 11-19). We also added the related discussion of these new results to the revised manuscript accordingly.

We followed your suggestion and removed the word “green”.

Concerning the application of polypeptides in biomedical fields, we have added descriptions and cited related literature in the revised MS. Polypeptides derived from naturally occurring α -amino acids is a unique and versatile family of bio-inspired biomaterials that can be tailor-made for varying biomedical applications, such as controlled drug release, gene delivery, tissue engineering and regenerative medicine (*Progress in Polymer Science*, 2014, 39, 330; *Chem. Commun.*, 2014, 50, 139.). In contrast to traditional biodegradable polymers such as aliphatic polyesters and polycarbonates, polypeptides are unique, allow precise control over polarity and charges, show excellent stability against hydrolysis, and are prone to rapid biodegradation *in vivo* by specific enzymes.

Regarding the catalyst residue in resultant polypeptides, we have checked the catalyst residue in the polymer samples by using ^1H NMR, ^{19}F NMR and EDS. We purposely prepared a high molecular weight polypeptide sample from polymerization with high loading of catalyst ($[\text{Glu-NCA}]/[\text{DMEA}]/[\text{1,3-bis-HFAB}]=120/0.25/10$, 100% monomer conversion). When the polymerization was completed, and the resultant polypeptide was precipitated using methanol, a further two-step washing process of the precipitated polypeptide with methanol can totally remove the catalyst from the final polymer. The ^1H NMR, ^{19}F NMR and EDS spectra show no **1,3-bis-HFAB** catalyst residue in the final polypeptide (Figures S13, S17, S18 and S19).

Furthermore, we also evaluated the cytotoxicity of the final polymer by doing *in vitro* cell experiments as follows. Briefly, HepG2 cells were seeded in 96-well plates with a density of 5×10^3 cells per well and incubated overnight. Subsequently, the cells were incubated for 48 h with various concentrations of the extract of the final polypeptides (the preparation of extract of polypeptide follows international standard ISO 10993-12:2012). Afterward, the cytotoxicity was evaluated by adding CCK-8

solution to each well of the plate. After incubation for 1 h, the absorbance was investigated at 450 nm using a Multiskan MK3 Microplate Reader. The final polypeptide did not show any cytotoxicity to HepG2 cells at the concentration below 0.2 mL/mL after 48 h, which was reflected by the fact that the cell viability still exceeded 95% (Figure S20). Also, by using live/dead cell staining technique, stained live and dead cells can be visualized by fluorescence microscopy (Figure S21). From fluorescence microscopy images, we can easily find that the viability of the cells with polypeptide extracting liquid was comparable to a blank sample (without polypeptide extracting liquid). These results suggest that the obtained polymer showed no cytotoxicity.

We have included these results and discussion in the revised MS (SI).

Figure S13 The combined ¹H NMR spectra of 1,3-bis-HFAB catalyst and PBLG from polymerization initiated by DMEA/1,3-Bis-HFAB (500M, 25°C, CDCl₃/CF₃COOD (2:1), (1) 1,3-bis-HFAB catalyst; (2) before precipitation with methanol; (3) the precipitated PBLG was further washed with methanol twice)

Figure S17 The combined ^{19}F NMR spectra of **1,3-bis-HFAB** catalyst and PBLG from polymerization initiated by **DMEA/1,3-Bis-HFAB** (500M, 25°C, CDCl_3), (1) 1,3-bis-HFAB catalyst; (2) before precipitation with methanol; (3) the precipitated PBLG was further washed with methanol twice)

Figure S18 EDS spectrum of PBLG from polymerization initiated by **DMEA/1,3-Bis-HFAB** (before precipitation with methanol)

Figure S19 EDS spectrum of PBLG from polymerization initiated by DMEA/1,3-Bis-HFAB (the precipitated PBLG was further washed with methanol twice)

Figure S20 In vitro cytotoxicity evaluation of the final polypeptide by the viability of HepG2 cells cultured for 48 h with various concentrations of extract from the final the polypeptide

Figure S21 Fluorescence microscopy images of HepG2 cells cultured for 48 h with various concentrations of the extracting liquid of the final polypeptides (I, in bright-field; II, images of cells observed in dark-field with $\lambda_{\text{excitation}}=488\text{nm}$, green color represents living cell and red for dead cell; III, images of dead cells observed in dark-field with $\lambda_{\text{excitation}}=552\text{nm}$). (Note: Identification of live and dead cells is very

important for the investigation of growth control and cell death. The Live-Dead Cell Staining Kit can provide ready-to-use reagents for convenient discrimination between live and dead cells. The kit utilizes Live-Dye™, a cell-permeable green fluorescent dye (Ex/Em = 488/518 nm), to stain live cells. Dead cells can be easily stained by propidium iodide (PI), a cell non-permeable red fluorescent dye (Ex/Em = 488/615). Stained live and dead cells can be visualized by fluorescence microscopy using a band-pass filter.

Reviewer #3

This paper report the controlled or living ring opening polymerization of NCA using aminoalcohol (DMEA) as the initiator, and a small molecular fluorinated alcohol, 1,3-bis-HFAB, as an organocatalyst. Through analysis of molecular structure of the polymer product and measuring polymerization kinetics, the authors proved that the ring opening polymerization was living and the chain growth was mediated by the reversible activation-deactivation of the initiating hydroxyl group and the propagating terminal amine group. A thorough study on NMR spectroscopy of the model compounds indicated that the activation-deactivation process was fulfilled through labile hydrogen bonding between HFAB and the corresponding species. In my opinion, the results in the present manuscript represent a remarkable progress in ring opening polymerization of NCA because the initiating/catalytic system is all organic and the polymerization is of high selectivity. On this basis, I recommend publication of the work by Nature Communication. The article was well written and the conclusion was logically supported by the results. I only find a few typos, as follows, which need minor revision before publication,

1) Line 100: Why you say the experimental M_n,SEC was calculated based on NMR? As far as I understand, the molecular weight should be $M_n,SEC-LS$.

Response:

Thank you for your comment.

It should be " $M_{n,NMR}$ ", because the molecular weight was calculated from the 1H NMR spectrum of the oligomer based on the signals of the **DMEA** initiator anchored at the polymer chain-end. We have changed " $M_{n,SEC}$ " into " $M_{n,NMR}$ " and marked in red in the revised MS.

2) Line 140 & 216: Here should be MALDI-TOF MS spectrum.

Response:

We have changed “MALDI-TOF spectra” into “MALDI-TOF MS spectrum” and marked the changes in red in the revised MS.

3) Line 169: The dependence of what? The sentence must be revised to be complete.

Response:

We have completed the sentence as: “The dependence of polymerization rate on initiator concentration was also determined.”, which is marked in red in the revised MS.

REVIEWERS' COMMENTS:

Reviewer #1 (Remarks to the Author):

The revisions including additional experiments are satisfactory.

Reviewer #2 (Remarks to the Author):

Thank you to the authors for making many needed changes to this manuscript. I think it is in much better form than its previous submission and should be accepted for publication.

Olivier Coulembier

REVIEWERS' COMMENTS:

Reviewer #1 (Remarks to the Author):

The revisions including additional experiments are satisfactory.

Reviewer #2 (Remarks to the Author):

Thank you to the authors for making many needed changes to this manuscript. I think it is in much better form than its previous submission and should be accepted for publication.

Olivier Coulembier

Point-by-Point Response to the reviewers' comments:

Reviewer #1 (Remarks to the Author):

The revisions including additional experiments are satisfactory.

Response:

We sincerely thank the reviewer for taking the time to evaluate our work and support from the reviewer.

Reviewer #2 (Remarks to the Author):

Thank you to the authors for making many needed changes to this manuscript. I think it is in much better form than its previous submission and should be accepted for publication.

Olivier Coulembier

Response:

We sincerely thank the reviewer for taking the time to evaluate our work and support from the reviewer.